# An iron (II) dependent oxygenase performs the last missing step of plant lysine catabolism

Mitchell G. Thompson [1,2,3,15], Jacquelyn M. Blake-Hedges[1,2,4,15], Jose Henrique Pereira[1,5,15], John A. Hangasky [4], Michael S. Belcher [1,2,3], William M. Moore[1,2,3], Jesus F. Barajas[1,2,6], Pablo Cruz-Morales[1,2], Lorenzo J. Washington[1,2,3], Robert W. Haushalter[1,2], Christopher B. Eiben[1,2,7], Yuzhong Liu[1,2], Will Skyrud[4], Veronica T. Benites[1,2], Tyler P. Barnum[3], Edward E. K. Baidoo[1,2], Henrik V. Scheller [1,2,3], Michael A. Marletta [4,8], Patrick M. Shih[1,2,9,10,11], Paul D. Adams[1,5,7] & Jay D. Keasling [1,2,7,12,13,14 ✉]

Despite intensive study, plant lysine catabolism beyond the 2-oxoadipate (2OA) intermediate remains unvalidated. Recently we described a missing step in the D-lysine catabolism of *Pseudomonas putida* in which 2OA is converted to D-2-hydroxyglutarate (2HG) via hydroxyglutarate synthase (HglS), a DUF1338 family protein. Here we solve the structure of HglS to 1.1 Å resolution in substrate-free form and in complex with 2OA. We propose a successive decarboxylation and intramolecular hydroxylation mechanism forming 2HG in a Fe(II)- and $O_2$-dependent manner. Specificity is mediated by a single arginine, highly conserved across most DUF1338 proteins. An *Arabidopsis thaliana* HglS homolog coexpresses with known lysine catabolism enzymes, and mutants show phenotypes consistent with disrupted lysine catabolism. Structural and biochemical analysis of *Oryza sativa* homolog FLO7 reveals identical activity to HglS despite low sequence identity. Our results suggest DUF1338-containing enzymes catalyze the same biochemical reaction, exerting the same physiological function across bacteria and eukaryotes.

[1] Joint BioEnergy Institute, Emeryville, CA, USA. [2] Biological Systems & Engineering Division, Lawrence Berkeley National Laboratory, Berkeley, CA, USA. [3] Department of Plant and Microbial Biology, University of California-Berkeley, Berkeley, CA, USA. [4] Department of Chemistry, University of California-Berkeley, Berkeley, CA, USA. [5] Molecular Biophysics and Integrated Bioimaging, Lawrence Berkeley National Laboratory, Berkeley, CA, USA. [6] Department of Energy Agile BioFoundry, Emeryville, CA, USA. [7] Department of Bioengineering, University of California-Berkeley, Berkeley, CA 94720, USA. [8] Department of Molecular and Cellular Biology, University of California-Berkeley, Berkeley, CA, USA. [9] Department of Plant Biology, University of California-Davis, Davis, CA, USA. [10] Genome Center, University of California-Davis, Davis, CA, USA. [11] Environmental Genomics and Systems Biology Division, Lawrence Berkeley National Laboratory, Berkeley, CA, USA. [12] Department of Chemical and Biomolecular Engineering, University of California-Berkeley, Berkeley, CA, USA. [13] The Novo Nordisk Foundation Center for Biosustainability, Technical University of Denmark, Lyngby, Denmark. [14] Center for Synthetic Biochemistry, Shenzhen Institutes for Advanced Technologies, Shenzhen, China. [15] These authors contributed equally: Mitchell G. Thompson, Jacquelyn M. Blake-Hedges, Jose Henrique Pereira. ✉email: jdkeasling@lbl.gov

Lysine is an essential amino acid, and due to its low abundance in cereals and legumes it is produced on a scale of one million tons a year to supplement food supply needs[1–3]. To thwart malnutrition in the developing world, significant work has been done to engineer rice, maize and other plants to produce greater quantities of lysine[3,4]. Increasing lysine levels in cereal grains requires overexpression of lysine-producing enzymes and concurrent disruption of lysine catabolism[3]. Thus, mutants such as *opaque2* in maize have received considerable attention for their ability to accumulate lysine within their endosperm[5,6]. However, despite worldwide importance, the full plant lysine catabolism pathway remains unknown, with no consensus in the steps beyond 2-oxoadipate (2OA) formation[7].

Recently we described a novel D-lysine catabolic route in the bacterium *Pseudomonas putida* which also contains a 2OA intermediate, similar to plant L-lysine catabolism[8]. In the *P. putida* pathway, 2OA is converted to 2-oxoglutarate (2OG) via three enzymes, one of which catalyzes a unique decarboxylation–hydroxylation step. In this step, 2OA is converted to D-2-hydroxyglutarate (2HG) in a reaction catalyzed by the Fe(II)-dependent DUF1338 family enzyme hydroxyglutarate synthase (HglS) (Supplementary Fig. 1)[8]. Homologs of this enzyme are broadly distributed across multiple domains of life, including nearly every sequenced plant genome[8,9]. However, only one study describing DUF1338 enzymes in plants has been reported. In *Oryza sativa FLO7* (a DUF1338 family member) mutants, abnormal starch formation was observed in the endosperm similar to maize *opaque2* mutants that are known to accumulate high levels of lysine[9]. The widespread DUF1338 family abundance in plants and the *FLO7* and *opaque2* phenotypes encouraged us to further investigate whether enzymes throughout the family displayed similar activity to HglS.

Conversion of 2OA directly to D-2HG requires two discrete chemical steps, a decarboxylation and hydroxylation, making the chemical mechanism of the enzyme puzzling[8]. Here, we leverage structural and biochemical analyses to postulate a chemical mechanism for HglS and characterize its substrate specificity. We show that critical residues involved in catalysis are highly conserved across nearly all DUF1338 family proteins. We further show that despite very low sequence identity, plant homologs also catalyze the conversion of 2OA to 2HG and adopt the same structural fold as the bacterial enzyme, suggesting that DUF1338 family proteins catalyze the last unknown step of plant lysine catabolism.

## Results

**Structural analysis reveals the catalytic mechanism of HglS.** To better understand the unusual HglS reaction, we obtained crystal structures of the enzyme both with and without a bound substrate. Initially, Hg1S was crystallized without substrate, and the Hg1S structure was solved at 1.1 Å resolution (Fig. 1a). The enzyme possesses a central β-sheet motif resembling a partially-closed β-barrel consisting of seven β-sheets, which is conserved in the three other DUF1338 structures deposited in the Protein Data Bank (PDB IDs 3LHO, 3IUZ, and 2RJB). Further analysis also revealed the presence of a metal cofactor bound within a conserved metal cofactor-binding motif—consisting of the residues His 70, His 226, and Glu 294—common to the DUF1338 structures. However, none of the available DUF1338 structures have been biochemically characterized or solved in complex with a substrate, and consequently no chemical reaction mechanism for the family has been proposed.

Therefore, we performed a search for characterized proteins with similar structures using the Vector Alignment Search Tool (VAST)[10]. The top VAST hits were the three reported DUF1338 structures,

followed by the hydroxymandelate synthase (HMS) and 4-hydroxyphenyl pyruvate dioxygenase (HPPD) structures (Supplementary Data 1)[11,12]. HglS and HMS structure comparison revealed the two enzymes share a similar central β-sheet fold common to the DUF1338 structures (Supplementary Fig. 2)[11,13]. More importantly, the β-sheet domain of HMS contains the enzyme active site, two histidines and a glutamate that bind the metal cofactor in nearly the same orientation as in HglS. The similarity of the HglS, HMS, and HPPD folds and active sites suggested that HglS is likely an additional member of the vicinal oxygen chelate (VOC) enzyme superfamily and could act via a similar mechanism to HMS and HPPD[14,15].

To determine metal cofactor identity within the HglS structure, we conducted a fluorescent scan of a HglS crystal. We detected a photon emission near 7475 eV, the K-alpha emission of nickel (Supplementary Fig. 3). In addition, a less intense peak corresponding to the K-alpha emission of iron was also observed. We therefore assigned the co-crystallized metal as Ni(II) with a small percent Fe(II) occupancy. While the metal cofactors present in the HMS and HglS structures differ, we previously determined that HglS utilizes Fe(II), not Ni(II), for catalysis. The nickel bound to HglS is likely derived from the nickel affinity chromatography protein purification.

The HglS domain architecture suggests the enzyme belongs to the VOC superfamily, mandating a chemical mechanism employing the bidentate coordination of vicinal oxygen atoms to a divalent metal center. Given the established HMS substrate-binding mode, we hypothesized that the metal in HglS would bind the vicinal oxygen atoms of the α-keto group of 2OA[11]. We therefore soaked HglS crystals with 2OA and solved the structure of the resulting complex. The noncatalytic active site nickel prevented enzyme turnover, yielding a substrate-bound structure, and no observed product density. As hypothesized, the substrate carboxylate and α-keto group oxygens are coordinated to the metal (Fig. 1b), classifying HglS as a VOC superfamily member.

We had previously noted the similarity between the set of reactions catalyzed by HglS and HMS[8,11,13]. Both enzymes perform a decarboxylation–hydroxylation reaction on a α-ketoacid substrate. HPPD catalyzes a similar overall reaction with the hydroxylation occurring at a different position. The structural and biochemical similarity of HglS to HMS and HPPD led us to propose a similar chemical mechanism for HglS (Fig. 1c). More specifically, the catalytic cycle begins when the substrate carboxylate and α-keto group oxygens and molecular oxygen bind to the three open Fe(II) coordination sites. A radical rearrangement results in a decarboxylation and the formation of a Fe(IV)-oxo and an α-radical species. Finally, continued radical rearrangement produces the hydroxylated product 2-hydroxyglutarate. If HglS follows this mechanism, it should consume 1 mol of $O_2$ per mole of 2OA, and two oxygens in the product should derive from molecular $O_2$. To support this mechanism, we determined the stoichiometry of 2OA to $O_2$ consumption using a dissolved oxygen probe. In the presence of 100 or 200 μM 2OA, the enzyme consumed approximately an equimolar concentration of $O_2$ (Fig. 1d). No feedback inhibition was observed with 1 mM D-2HG, or L-2HG (Supplementary Fig. 4a), and kinetic parameters determined by monitoring oxygen consumption were similar to those we previously reported using a different enzyme assay (Supplementary Fig. 4b)[8]. We next tested our hypothesis that two oxygen atoms in the 2HG product derive from molecular $O_2$. To determine the source of oxygen atoms, additional enzyme assays were performed under an $^{18}O_2$ atmosphere. High resolution LC-MS product analysis revealed a species exhibiting a *m/z* of 151.022 that co-eluted with a 2HG standard (*m/z* 147.029), corresponding to the expected *m/z* of 2HG containing two $^{18}O$-labeled oxygens (Fig. 1e). These results strongly suggest HglS proceeds via a mechanism similar to HMS and is

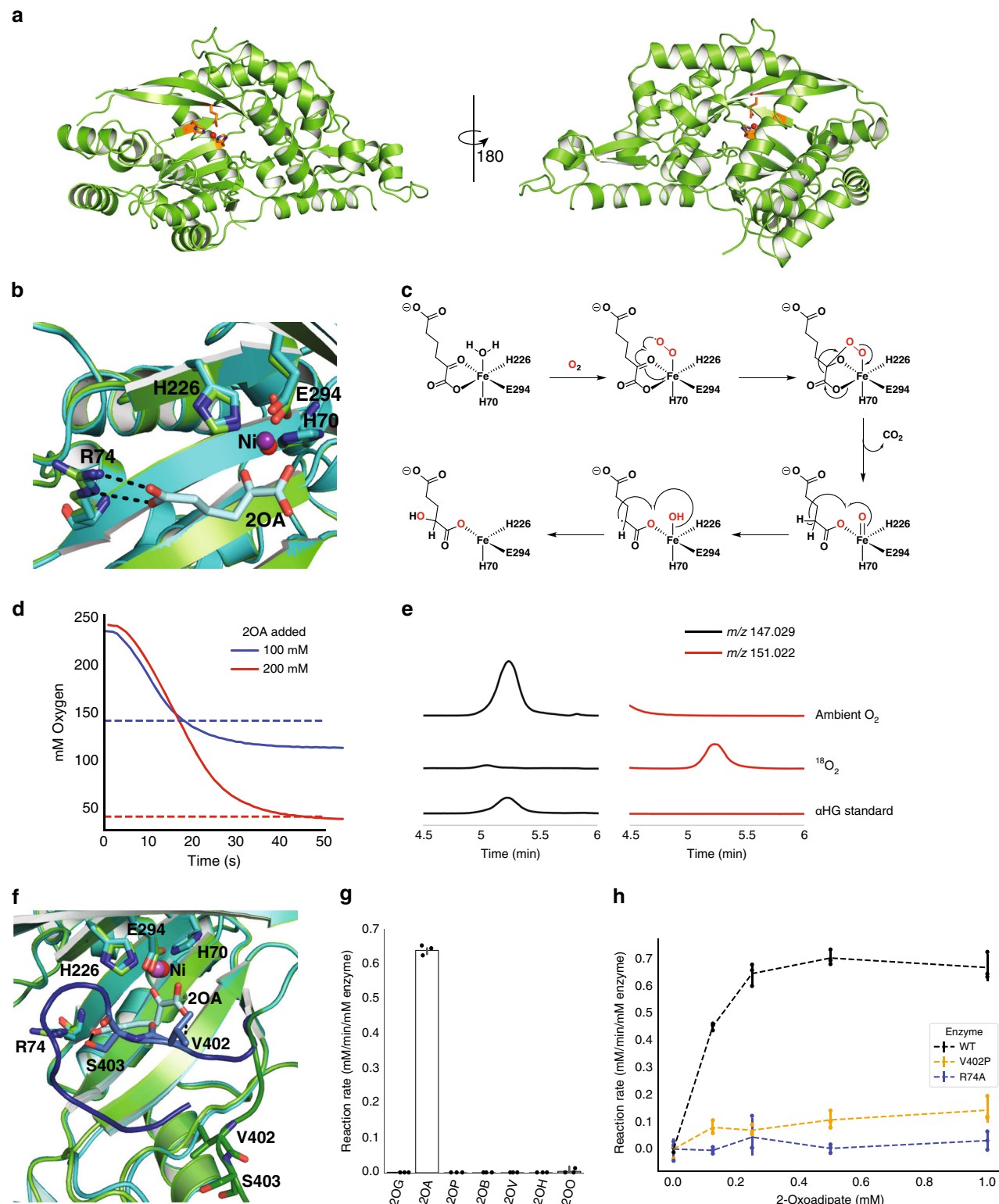

therefore the third identified member of the two-substrate α-ketoacid-dependent oxygenase (2S-αKAO) family[14].

Further analysis of the co-crystal structure revealed several other substrate-binding residues. Specifically, arginine 74 forms a salt bridge with the distal carboxylate of the 2OA substrate. In addition, valine 402 and serine 403 form hydrogen bonds with the carboxylate keto group and distal carboxylate group, respectively (Fig. 1f). The loop containing residues 402 and 403 shifts

approximately 10 Å from the *holo* state to the enzyme-bound state, bringing it into proximity with the substrate. We investigated the importance of Arg74 in determining enzyme specificity by assaying enzyme activity against a panel of α-ketoacids (Fig. 1g). HglS exhibited activity only with 2OA, and displayed no detectable and statistically significant activity above background (a boiled enzyme control) on any other substrate. We therefore concluded that Arg74 participates in favorable

**Fig. 1 Structural and Biochemical Analyses of *P. putida* HglS. a** Ribbon diagram of the HglS crystal structure. The left image shows the active site entrance, containing the metal cofactor (red sphere) and metal-coordinating residues (orange) located in the central β-sheet protein domain. A 180° rotation on the right shows the overall protein structure. **b** Overlay of *holo* (cyan) and substrate-bound (green) structures of HglS displaying the enzyme active site and the ionic interaction between Arg74 and the distal carboxylate group of the 2OA substrate. The nickel bound within each structure is displayed as a red (*holo* structure) or purple (substrate-bound structure) sphere. **c** Proposed reaction mechanism of HglS. $O_2$-derived oxygens are shown in red. **d** Oxygen to 2OA stoichiometry of the HglS reaction. Dissolved oxygen concentration (*y*-axis) was measured with a Clark-type oxygen probe. Either 200 μM (red line) or 100 μM (blue line) of 2OA was added to initiate reaction, resulting in equimolar oxygen consumption. Dotted lines represent the expected final $O_2$ concentration when f100 μM (blue line) or 200 μM (red line) 2OA is consumed. **e** LC-HRMS extracted ion chromatograms (EICs) showing labeled $^{18}O$ incorporation into the 2-hydroxyglutarate product of HglS. On the left (black lines) are EICs for ions with *m/z* 147.029, representing 2-hydroxyglutarate containing $^{16}O$. On the right (red lines) are EICs for ions with *m/z* 151.022, representing 2-hydroxyglutarate containing two $^{18}O$ atoms. A control reaction performed under ambient $O_2$ conditions (*n* = 3) is compared with a reaction performed under a $^{18}O_2$ atmosphere (*n* = 3) and a 2-hydroxyglutarate standard. **f** Overlay of *holo* (green) and substrate-bound (cyan) structures of HglS displaying the enzyme active site and the interaction of Val402 and Ser403 with the 2-oxoadipate substrate. The loop containing Val402 and Ser403 is shown in the *holo* (dark green) and substrate-bound (dark blue) states. **g** Reaction rates of HglS with different 2-oxoacid substrates: 2-oxoglutarate (2OG), 2-oxoadipate (2OA), 2-oxopimelate (2OP), 2-oxobutyrate (2OB), 2-oxovalerate (2OV), 2-oxohexanoate (2OH), 2-oxooctanoate (2OO). Error bars represent 95% confidence intervals, black dots represent individual measurements, *n* = 3. **h** Reaction rates of WT HglS (black dashes), R74A (blue dashes), and V402P (orange dashes) mutants measured by an enzyme coupled decarboxylation assay with 2OA as a substrate, *n* = 3. Error bars represent 95% confidence intervals, colored dots correspond to individual measurements.

substrate-binding interactions, but it also appears to influence the strict enzyme substrate specificity. Furthermore, an R74A mutation abolished enzymatic activity (Fig. 1h). We additionally probed the importance of the loop bearing Val402 and Ser403 on substrate binding. Mutating Val402 to a proline residue to disrupt the hydrogen bond observed between 2OA and the amide backbone of Val402 significantly decreased enzyme activity, suggesting that while this residue is not essential for turnover, it likely contributes to substrate-binding affinity (Fig. 1h). However, it is also possible that the observed decrease in activity is an artefact of disrupted protein folding due to the introduction of a proline residue. Further experiments with more conservative mutations at positions 402 and 403 will be required to fully understand the role(s) of these residues in substrate binding and/or catalysis.

**DUF1338 biochemistry and structure is conserved in homologs**. Based on the mechanistic information gleaned from our HglS biochemical and structural characterization, we sought to propose potential functions for other DUF1338 family members. Previously, we showed that DUF1338 family proteins are widely distributed across several domains of life, while others have demonstrated that DUF1338 protein coding sequences are present in the majority of sequenced plant genomes[8,9]. In plants, the catabolism of lysine is known only until 2OA, with further catabolic steps having only been hypothesized[7]. Furthermore, D-2HG has also been identified as an intermediate in plant lysine catabolism, though the mechanism of its formation has yet to be proven. We therefore hypothesized that plant homologs perform the same reaction as HglS, converting 2OA to 2HG. To test this, we biochemically characterized the DUF1338 homologs from *Arabidopsis thaliana* and *O. sativa* as well as an additional distantly related *Escherichia coli* homolog. Soluble variants of plant proteins were constructed by removing the predicted N-terminal localization peptide. As expected, the *E. coli* homolog YdcJ and the plant homologs AT1G07040 and FLO7 catalyzed the conversion of 2OA to that was confirmed by in vitro assays analyzed using high resolution LC-MS (Supplementary Fig. 5a). In addition, the FLO7 homolog displayed kinetic parameters similar to HglS, with a $K_m$ of 0.55 mM and a $V_{max}$ of 0.89 mM/min/μM enzyme (Supplementary Fig. 5b).

In addition, the structural conservation between bacterial and eukaryotic DUF1338 proteins was compared by obtaining a crystal structure of FLO7. Initial crystallization screens resulted in crystals that were soaked with 2OA, producing diffraction data

used to solve the substrate-bound crystal structure at 1.85 Å resolution. The FLO7 crystal structure, like the other DUF1338 proteins, displayed the conserved central VOC fold containing the active site and metal-binding center (Fig. 2a). Comparison of the HglS and FLO7 structures revealed that orientation of the metal-coordinating residues and 2OA in both structures were nearly identical (Fig. 2b) even though the proteins display low (~15%) sequence identity (Supplementary Fig. 6). In addition, the FLO7 structure also contains a substrate-binding arginine, Arg64, located in nearly the same position as Arg74 of HglS. However, unlike Val402 and Ser403 of HglS, FLO7 does not possess any other residues interacting with the substrate. The results of our biochemical and structural studies of FLO7 manifest the remarkable conservation of the fold, active site architecture, and mechanism among DUF1338 proteins across different domains of life. Consequently, we predict that all DUF1338 proteins containing the conserved 2OA-interacting arginine likely catalyze the decarboxylation and hydroxylation of 2OA to form 2HG. These hydroxyglutarate synthases form a new two-substrate α-ketoacid-dependent oxygenase (2S-αKAO) subfamily lacking significant sequence identity to HMS or 4-HPPD and acting on a distinct substrate.

**Conserved residues suggest a common biochemical role**. The conserved structural features, substrate, and biochemical activity of plant and bacterial DUF1338 proteins suggest a common physiological role across domains of life. Therefore we analyzed all known DUF1338 domain-containing protein amino acid sequences for the key catalytic and substrate-binding residues we identified in HglS. First, we queried all DUF1338 proteins within the Pfam database for the catalytic and metal-binding residues identified in HglS. Of the 2417 unique DUF1338-containing proteins found in the Pfam PF07063 family, 86% possess the conserved "HHE" metal-binding triad (Supplementary Table 1). This strong conservation is consistent across domains of life, with 85% of DUF1338 proteins in bacteria, 92% in fungi, and 81% in plants possessing the metal-binding residues. Further analysis revealed arginine 74, which dictates HglS substrate specificity, is also highly conserved across the family. Of the homologs with conserved HHE triads, 100% of fungal proteins, 99.7% of bacterial proteins, and 93.4% of plant proteins maintain an arginine at this position (Supplementary Table 1). Therefore, though there is little primary sequence identity between bacterial homologs and plant homologs (Supplementary Figs. 6 and 7), the critical residues coordinating the Fe(II) cofactor and the carboxylate-coordinating

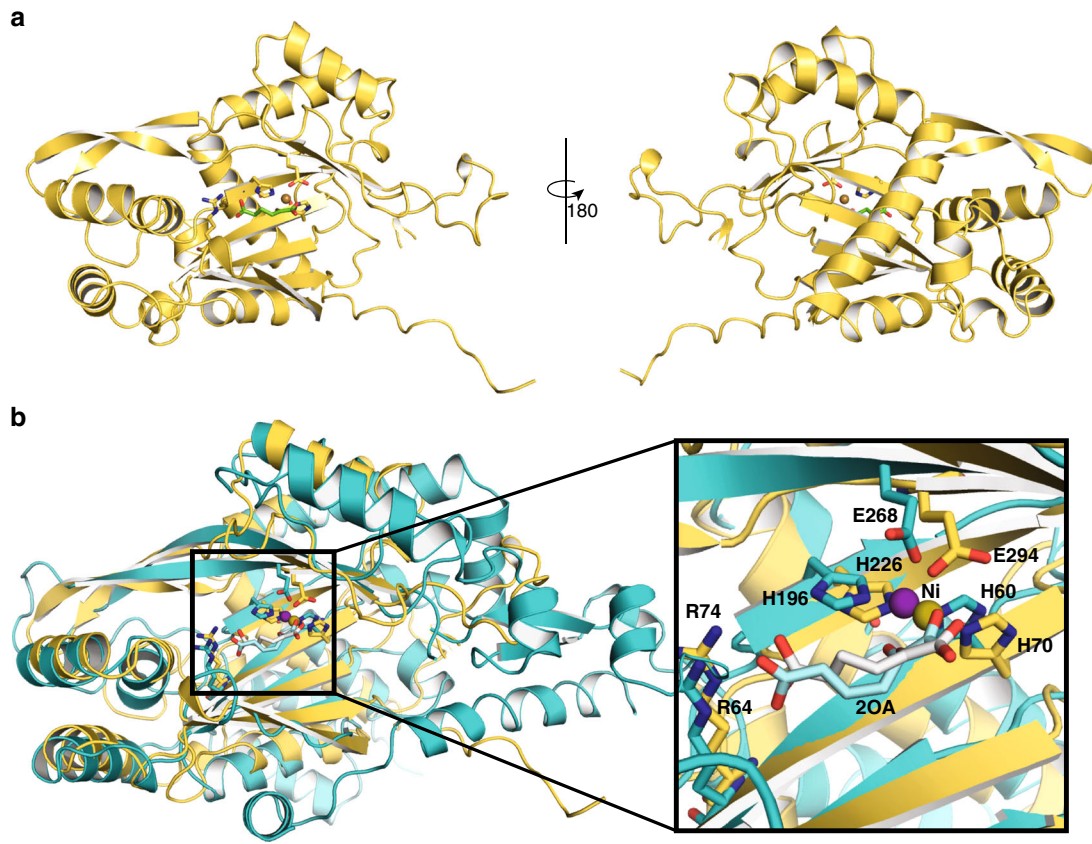

**Fig. 2 Structural comparison of FLO7 and HglS. a** FLO7 crystal structure displayed as cartoon sheets and helices. Metal-coordinating and substrate-binding residues are shown as sticks, while the 2-oxoadipate substrate is shown as green sticks. The nickel bound within the active site is shown as a tan sphere. **b** Overlay of HglS (teal) and FLO7 (yellow) structures showing metal-coordinating residues, substrate-binding residues, the active site metal, and the 2-oxoadipate substrate. Right inset: HglS (teal) and FLO7 (yellow) active sites displaying conserved residues and substrate-binding mode. The 2OA substrates are colored light blue (HglS) and white (FLO7), and the active site bound nickel shown as spheres and colored blue (HglS) and olive (FLO7).

arginine are highly conserved. Given the substrate specificity exhibited by the *P. putida* DUF1338 protein HglS, we hypothesized that the remaining uncharacterized homologs maintaining the metal-binding residues and conserved arginine would also use 2OA as a substrate.

Nearly all plant and algal genomes within GenBank encode DUF1338 family proteins (Supplementary Table 1). Previous work to understand plant lysine catabolism suggested the missing reactions between 2OA and 2HG proceed through the 2-ketoglutarate dehydrogenase complex (2KGD), forming glutaryl-Coenzyme A (Supplementary Fig. 8)[7]. Transcript correlation analysis in *A. thaliana*, however, revealed that the transcription of known lysine catabolic enzymes was not correlated with 2KGD expression; rather, their transcription is highly correlated with a DUF1338 protein AT1G07040 (Supplementary Table 2). In addition, in *O. sativa*, mutants of the DUF1338 protein FLO7 display a floury starch phenotype that resembles the maize *opaque2* phenotype which is known to cause lysine accumulation in mutant kernels[3,5,6,9]. These transcriptomic and phenotypic correlations suggest that, like HglS in *P. putida*, DUF1338 proteins are involved in the plant lysine degradation pathway.

During seed maturation, lysine catabolism is required for normal amyloplast development[16]. In rice, *FLO7* localizes to the amyloplast via a N-terminal chloroplast localization signal peptide[9]. While the *A. thaliana* homolog AT1G07040 also has a predicted chloroplast localization sequence, only 117 of 202 plant proteins had predicted chloroplast or mitochondria localization tags (Supplementary Table 3). Many predicted non-localized

proteins are isoforms of loci that also have putatively localized isoforms, though phylogenetic analysis also revealed a distinct non-localized DUF1338 protein clade predominantly within *Brassicaceae* (Supplementary Fig. 9). Notably, this non-localized subclade was enriched for proteins maintaining the HHE coordination triad but that possessed either a glutamine or methionine at the position corresponding to Arg74 of HglS (Supplementary Fig. 9). As most plant homologs contain a localization tag, we predicted these homologs would likely function similar to rice *FLO7*.

We therefore tested whether mutants of the DUF1338 protein AT1G07040 from the model plant *A. thaliana* exhibited a phenotype resembling other plant lysine-accumulating mutants. When grown on germination media, Salk_103299C mutants (which have confirmed homozygous T-DNA insertions in the AT1G07040 locus) displayed significantly delayed germination compared with wild-type seedlings (Supplementary Fig. 10a). AT1G07040 mutants that germinated also displayed compromised development and appeared cholortic with impaired growth (Supplementary Fig. 10a). This phenotype was recapitulated when seeds were germinated in soil. After a 14 day incubation, no Salk_103299C seeds germinated whereas 95% of wild-type seeds had germinated ($n = 96$) (Supplementary Fig. 10c). After 45 days, only 7% of mutant seeds had germinated and after 60 days, only 15%. This phenotype is consistent with previous work showing lysine accumulation in *A. thaliana* seeds resulted in significantly delayed germination[16,17]. Toluidine blue staining of sectioned mutant seeds also suggested that oil body formation was altered (Supplementary Fig. 10b). The altered morphology of the oil

body, serving as the primary carbon storage unit in *A. thaliana*, is analogous to the altered starch granule formation observed in rice *FLO7* mutants[9]. The aberrant phenotype in both *A. thaliana* and rice mutants suggest that DUF1338 homologs play a critical role in embryo development both in monocots and dicots. The phenotypes in *A. thaliana* observed in this preliminary work will need to be validated using multiple confirmed T-DNA lines that disrupt the function of AT1G07040.

## Discussion

Here, we present the structural and mechanistic analysis of HglS, the first DUF1338 protein family member to be assigned a function. In a previous report, we showed that HglS converts 2OA to 2HG. The HglS chemical mechanism remained ambiguous however, as this conversion involves two apparent enzymatic steps: decarboxylation and hydroxylation. Here we report the high resolution crystal structures of holo- and substrate-bound enzyme aiding the proposal of an enzyme mechanism. In addition, we present the substrate-bound structure of the plant homolog, FLO7.

Several HglS unique features were used to propose an enzymatic reaction mechanism. A VAST structural similarity search revealed the central VOC superfamily β-sheet fold, and metal-binding residues of HglS were positioned similarly to the corresponding HMS and HPPD residues (Supplementary Data 1). Both HMS and HPPD catalyze an intramolecular decarboxylation–hydroxylation reaction but share little sequence identity with HglS[10]. Furthermore, a 2OA-bound HglS co-crystal structure revealed bidentate coordination of the 2OA α-keto group by the metal center as a key step in the enzymatic mechanism. These features were remarkably conserved in the FLO7 co-crystal structure, while the α-helices surrounding the central domain diverge.

The VOC fold is conserved across the protein superfamily, yet the specific orientation of β-sheets in HglS, FLO7, HMS, and HPPD appears to be most conserved in the intramolecular 2S-αKAO subfamily. With only seven β-sheets composing the central barrel-like domain, the HglS structure diverges slightly from the canonical VOC fold which contains eight β-sheets.

The central VOC domains of HglS, FLO7, HMS, and HPPD are distinct from that of the mechanistically similar α-ketoglutarate (αKG) dependent dioxygenases which possess a double-stranded β-helix (jelly roll) fold containing the metal-binding center and active site[18]. However, the orientation of metal-binding residues and bound substrate is understandably homologous given the similar chemical mechanisms of the enzyme families, which we verified through several biochemical experiments. Whether the shared general mechanism of the 2S-αKAOs and αKG-dependent dioxygenases is a product of convergent evolution remains unclear.

While HMS and HPPD show significant sequence identity and share the same substrate, the DUF1338 family proteins are not homologous to HMS or HPPD, nor do they share significant sequence identity with each other (~15% identity between HglS and FLO7). Low sequence conservation is a general VOC superfamily feature, but the low sequence identity between members catalyzing the same reaction is less common[14]. Despite low primary sequence conservation, the biochemical assays of mutants, structural analysis, and bioinformatics reported herein show that very few residues are essential for enzyme turnover in the hydroxyglutarate synthase family. It is possible that the α-helices surrounding the central VOC domain have been selected for other attributes, such as mediating protein–protein interactions as observed in the αKG-dependent dioxygenases[18].

Previous work has shown DUF1338 family proteins are widespread in bacteria and fungi, but they are especially prevalent in plants[8,9]. Our work here shows that the arginine residue that mediates the 2OA specificity of HglS is highly conserved across these homologs. It is therefore likely that all DUF1338 proteins with the conserved arginine catalyze the decarboxylation and hydroxylation of 2OA to form 2HG. The enzyme's substrate, 2OA, is primarily known as a lysine catabolism intermediate, suggesting that DUF1338 enzymes participate in similar biochemical processes across domains of life (http://modelseed.org/biochem/compounds/cpd00269). Previous plant phenotypic studies support this hypothesis; a rice study showed DUF1338-containing protein FLO7 disruption produced a crystalline starch phenotype within the amyloplast[9], similar to the lysine-accumulating *opaque2* maize mutants[19].

DUF1338 protein AT1G07040 expression in *A. thaliana* is highly correlated with other known lysine catabolism enzymes. In this work, we further support this claim by showing that when AT1G07040 is disrupted, *A. thaliana* seedlings have significantly compromised germination ability. Previous work showed *A. thaliana* lysine-accumulating mutants have delayed germination rates due to unfavorable TCA cycle effects, suggesting that AT1G07040 could be involved in lysine degradation[16,17]. Furthermore, by assaying the *A. thaliana* and rice enzymes in vitro, we show that AT1G07040 and FLO7 catalyze the transformation of 2OA to 2HG. While both compounds were previously identified as plant lysine catabolism intermediates, no clear link between the two molecules had been demonstrated[20]. Finally, histopathological examination of mutant AT1G07040 seeds showed aberrant oil body formation. While previous experiments showed that the disruption of lysine utilization compromises the ability of seeds to store carbon in monocots such as rice and maize, our results show this phenomenon also occurs in dicots such as *Arabidopsis*. Given their near ubiquitous conservation and highly specific biochemical function we find it likely that DUF1338 proteins localized to chloroplasts catalyze the last missing step in lysine catabolism of all green plants.

While the majority of plant HglS homologs have chloroplast localization tags, some lack any predicted signal peptide. This is especially prevalent in the *Brassicaceae*, where many species appear to have non-localized HglS paralogs. The majority of these paralogs retain arginine as their specificity residue, but they may have roles in pathways beyond lysine catabolism as the *A. thaliana* paralog AT1G27030 (lacking a localization sequence) shows no expression correlation with known lysine catabolic genes (Supplementary Table 2).

Recent studies show that the lysine-derived intermediates pipecolate and N-hydroxy-pipecolate can initiate systemic acquired resistance (SAR) in plants[21,22]. SAR, a global response, grants lasting broad-spectrum disease protection in uninfected tissue[23]. In bacteria, pipecolate is often catabolized to 2OA, suggesting that HglS homologs should receive future attention when studying pipecolate metabolism in plants. In addition to the non-localized paralogs retaining the conserved arginine, multiple *Brassicaceae* paralogs have altered residues at this position, harboring either methionine or asparagine. These paralogs likely catalyze the decarboxylation and hydroxylation of substrates other than 2OA. Further research discerning the physiological and biochemical functionality of "mutant" paralogs is warranted.

DUF1338 proteins are also widely distributed in both *Ascomycota* and *Basidiomycota* fungi, though the model fungi *Saccharomyces cerevisiae* or *Neurospora crassa* lack homologs. Within the fungal homologs examined here, all proteins containing a HHE metal-binding triad also maintained the arginine specificity residue suggesting that all function on a 2OA substrate. Unfortunately, almost no fungal catabolic lysine pathways are fully characterized genetically or biochemically[24]. Multiple studies, however, suggested 2OA is a likely lysine catabolism intermediate in *Pyriculuria oryzae*

and *Candida albicans*[25,26]. Moreover, while *P. oryzae* possesses a DUF1338 homolog, *C. albicans* does not. This implies *C. albicans* and other fungi may utilize a catabolic route similar to mammals in which 2OA is converted to glutaryl-CoA via 2OA dehydrogenase[27]. Future studies are required to examine whether fungal HglS homologs also play a role in fungal lysine catabolism.

Of the over 2000 bacterial HglS homologs examined here that retain the HHE triad, over 99% maintained arginine as the specificity-conferring residue, suggesting widespread DUF1338 enzymatic activity conservation in prokaryotes. Previous work in *P. putida* and published fitness data in *Pseudomonas fluorescens* and *Sinorhizobium meliloti* provide evidence of a conserved physiological function as well[8,28,29]. *E. coli* was recently shown to possess a non-ketogenic lysine catabolic route via a glutarate hydroxylase, supplementing degradation routes to cadaverine[30,31]. *E. coli* also possesses a HglS homolog, YdcJ, which we showed has identical activity to HglS. In addition, the *E. coli* enzyme structure was solved (PDB ID 2RJB) and shows the same conserved VOC fold, metal-binding motif, and arginine-binding residues as the HglS and FLO7 structures reported herein. Future work will elucidate whether, like *P. putida*, *E. coli* possesses multiple lysine catabolism routes or whether 2OA functions in other physiological processes.

Low cereal and legume lysine content produces protein-energy malnutrition in 30% of the developing world population[3,32–35]. Increasing lysine content in staple crops will require both lysine overproduction and catabolic pathway elimination[34]. However, lysine metabolism changes in maize, rice, and soybeans resulted in low germination rates, abnormal endosperm, and reduced grain weights[3,19,36–38]. We hope that the more complete understanding of lysine catabolism elucidated here will help resolve causes of pleiotropic effect in plants and aid in the development of stable high-lysine crops to combat malnutrition globally.

## Methods

**Media, chemicals, and strains**. Routine bacterial cultures were grown in Luria-Bertani (LB) Miller medium (BD Biosciences, USA). *E. coli* was grown at 37 °C. Cultures were supplemented with carbenicillin (100 mg/L, Sigma Aldrich, USA). All compounds with the exception of 2-oxohexanoic acid were purchased through Sigma Aldrich. All bacterial strains and plasmids used in this work are listed in Supplementary Table 4 and are available through the public instance of the JBEI registry. (https://public-registry.jbei.org/). *A. thaliana* mutants were obtained from the Salk collection.

**DNA manipulation**. All plasmids were designed using Device Editor and Vector Editor software, while all primers used for the construction of plasmids were designed using j5 software[39–41]. All primers used in this study can be found in Supplementary Table 5. Plasmids were assembled via Gibson Assembly using standard protocols[42], or Golden Gate Assembly using standard protocols[43]. Plasmids were routinely isolated using the Qiaprep Spin Miniprep kit (Qiagen, USA), and all primers were purchased from Integrated DNA Technologies (IDT, Coralville, IA). Site directed mutants were created by incorporating desired mutations into PCR primers. PCR fragments were then re-assembled into the mutant plasmid using Golden Gate assembly[44]. The geneblock for the *E. coli* codon optimized *O. sativa* was purchased through IDT (Coralville, IA). Arabidopsis cDNA was used to amplify AT1G07040.

**Protein purification**. A 5 mL overnight culture of *E. coli* BL21 (DE3) containing the expression plasmid was used to inoculate a 500 mL culture of LB. Cells were grown at 37 °C to an OD of 0.6 then induced with Isopropyl β-D-1-thiogalactopyranoside to a final concentration of 1 mM. The temperature was lowered to 30 °C and cells were allowed to express for 18 h before being harvested via centrifugation. Cell pellets were stored at −80 °C until purification. For purification, cell pellets were resuspended in lysis buffer (50 mM sodium phosphate, 300 mM sodium chloride, 10 mM imidazole, 8% glycerol, pH 7.5) and sonicated to lyse cells. Insolubles were pelleted via centrifugation (30 min at 40,000 × *g*). The supernatant was applied to a fritted column containing Ni-NTA resin (Qiagen, USA), which had been pre-equilibrated with several column volumes of lysis buffer. The resin was washed with lysis buffer containing 50 mM imidazole, then the protein was eluted using a stepwise gradient of lysis buffer containing increasing

imidazole concentrations (100, 200, and 400 mM). Fractions were collected and analyzed via SDS-PAGE. Purified protein was dialyzed overnight at 4 °C against 50 mM HEPES pH 7.5, 5% glycerol.

**Crystallization**. An initial crystallization screen was set up using a Phoenix robot (Art Robbins Instruments, Sunnyvale, CA) using the sparse matrix screening method[45]. Purified HglS was concentrated to 20 mg/mL and Flo7 was concentrated to 10 mg/mL prior to crystallization using the sitting drop method in 0.4 μL drops containing a 1:1 ratio of protein sample to crystallization solution. For HglS, the crystallization solution consisted of 0.2 M Ammonium Fluoride and 20% PEG 3,350, while the crystallization solution for Flo7 contained 0.01 M Magnesium chloride hexahydrate, 0.05 M MES monohydrate pH 5.6 and 1.8 M Lithium sulfate monohydrate. Crystals were transferred to crystallization solution containing 20% glycerol prior to flash freezing in liquid nitrogen.

**X-ray data collection and model refinement**. X-ray diffraction data for HglS were collected at the Berkeley Center for Structural Biology on beamline 5.0.2 and 8.2.2 of the Advanced Light Source at Lawrence Berkeley National Lab. Diffraction data for Flo7 were collected at the Stanford Synchrotron Radiation Lightsource on beamline 12-2. The HglS and Flo7 structures were determined by the molecular-replacement method with the program *PHASER*[46] using uncharacterized protein YdcJ (SF1787) from *Shigella flexneri* (PDB: 2RJB) and the putative hydrolase (YP_751971.1) from *Shewanella frigidimarina* (PDB ID: 3LHO) as the search models, respectively. Structure refinement was performed by *phenix.refine* program[47]. Manual rebuilding using COOT[48] and the addition of water molecules allowed for construction of the final model. The R-work and R-free values for the final models of all structures are listed in Supplementary Table 6. Root-mean-square deviations from ideal geometries for bond lengths, angles, and dihedrals were calculated with Phenix[49]. The overall stereochemical quality of the final models was assessed using the MolProbity program[50]. Structural analyses were performed in Coot[48], PyMOL (https://pymol.org/2/)[51], and UCSF Chimera[52]. All structural data has been submitted to the Protein Database with the following PDB IDs: HglS: 6W1G, HglS-2OA: 6W1H, Flo7-2OA:6 W1K.

**Enzyme kinetics and O₂ consumption**. Enzyme coupled decarboxylation assays were carried out as previously described[53]. Reaction mixtures contained 100 mM Tris-HCl (pH 7), 10 mM MgCl₂, 0.4 mM NADH, 4 mM phosphoenol pyruvate (PEP), 100 U/mL pig heart malate dehydrogenase (Roche), 2 U/mL microbial PEP carboxylase (Sigma), and 10 mM 2OA. Reactions were initiated by the addition of purified HglS or boiled enzyme controls, and absorbance at 340 nm was measured via a SpectraMax M4 plate reader (Molecular Devices, USA).

Initial rate measurements were directly recorded monitoring the consumption of O₂ using a Clarke-type electrode (Hansatech Oxygraph). Reaction mixtures containing FeSO₄ (10 μM) and 2-oxoadipic acid (10–200 μM) in 100 mM Tris pH 7.0 were allowed to equilibrate to room temperature determined by a stable O₂ concentration reading of 240 μM. Addition of purified *apo* enzyme (100 nM) to the sealed reaction vial initiated the reaction. Initial rates were determined from the linear portion of consumption of O₂ corresponding to up to 10% consumption of the limiting reactant. No burst or lag phases were observed. All assays were performed in triplicate.

O₂ consumption measurements used to determine the reaction stoichiometry were also measured using a Clarke-type electrode. Reactions mixtures containing FeSO₄ (10 μM) and 2-oxoadipic acid (100 or 200 μM) in 100 mM Tris pH 7.0 were equilibrate to room temperature ([O₂] = 240 μM) and monitored for at least 2 min prior to the addition of *apo* enzyme (1 μM). Oxygen consumption was monitored until the signal plateaued and the O₂ concentration was stable. The observed rate was determined by fitting the data to a single exponential decay model. The reaction stoichiometry was determined by taking the ratio of the moles of O₂ consumed and the concentration of the 2-oxoadipic acid present in the reaction. Each reaction condition was performed in triplicate.

**Oxygen labeling experiments**. All reagents were exhaustively purged with argon on a Schlenk link to remove ¹⁶O₂. Anaerobic buffer was subsequently saturated with ¹⁸O₂ via gentle bubbling with ¹⁸O₂ (Sigma, 99 atom % ¹⁸O). Reaction mixtures containing enzyme (1 μM), 2-oxoadipic acid (1 mM), FeSO₄ (10 μM), and ¹⁸O₂ saturated buffer (1.2 mM) were mixed in anaerobic sealed reaction vials using gastight syringes. The reaction was initiated by the addition of 2-oxoadipic acid. The headspace was filled with ¹⁸O₂ gas. Reactions were incubated at room temperature for 2 h before being quenched with an equal volume of methanol. Control experiments, replacing ¹⁸O₂ with ¹⁶O₂, were run in parallel. Quenched reaction mixtures were analyzed by LC-MS.

**LC-MS analysis**. All in vitro reactions to be analyzed via LC-MS were quenched with an equal volume of ice cold methanol and stored at −80 °C until analyses. Detection of 2OA and 2HG were described previously[8]. Briefly, HILIC-HRMS analysis was performed using an Agilent Technologies 6510 Accurate-Mass Q-TOF LC-MS instrument using positive mode and an Atlantis HILIC Silica 5 μm

column (150 × 4.6 mm) with a linear of 95 to 50% acetonitrile (v/v) over 8 min in water with 40 mM ammonium formate, pH 4.5, at a flow rate of 1 mL min$^{-1}$.

**Plant growth**. *A. thaliana* DUF1338 mutant, Salk_103299C, seeds were ordered from the Arabidopsis Biological Resource Center (Columbus, OH, USA). Seeds were surface sterilized with 70% EtOH for 3 min, followed by a ten-minute sub-mersion in 10% bleach solution, then rinsed with sterile water three times. For soil germination, seeds were planted in Premier Pro-Mix mycorise pro soil, two seeds per well in a twenty-four well tray. Trays with seeds were then stratified at 4 °C for 3 days, then grown in a Percival-Scientific growth chambers at 22 °C in 10/14-h light/dark short-day cycles with 60% humidity. After the formation of the full rosette, plants were genotyped to confirm homozygosity for the mutation of interest. Primers were designed using the automated method from http://signal.salk.edu/cgi-bin/tdnaexpress, and genotyping was done in accordance with a pre-viously described protocol and diagrammed in Supplementary Fig. 11[54]. After confirmation of homozygous mutants, plants were moved into individual wells and transferred to long-day conditions, 22 °C in 16/8-h light/dark cycles with 60% humidity, to induce flowering for seed collection. For germination on synthetic media, sterilized seeds were plated on solid media supplemented with ½ Murashige and Skoog media base, 5% sucrose, 0.8% agar, and 10 μM gibberellic acid to induce germination. Plates with seeds were stratified at 4 °C for 3 days then transferred to a Percival-Scientific growth chambers at 22 °C in 10/14-h light/dark short-day cycles with 60% humidity.

**Synthesis of 2-oxohexanoic acid**. Synthesis of 2-oxohexanoic acid was carried out as described previously[55]. Briefly, a solution of the Grignard reagent, prepared from 1-bromobutane (500 mg, 3.68 mmol) and the suspension of magnesium (178 mg, 7.33 mmol) in THF (5 mL) was added dropwise under $N_2$ atmosphere to a solution of diethyloxalate (487 mg, 3.33 mmol) in THF (4 mL) at −78 °C. After the addition was complete, the reaction mixture was stirred at −78 °C for an additional 5 h. The reaction was quenched with 2 N HCl, the aqueous layer was extracted with ether, and the combined organic layer was washed with brine, dried over $MgSO_4$, and evaporated. The crude product was dissolved in acetic acid (20 mL) and conc. HCl (5 mL). After 11 h, the reaction was concentrated directly, and the residue was purified by distillation under reduced pressure to give the pure product (203 mg, 43%) as a colorless oil. $^1H$ NMR (400 MHz, $CDCl_3$) 2.92 (t, 2H), 1.81–1.54 (m, 2H), 1.48–1.28 (m, 2H), 0.92 (t, 3H) (Supplementary Fig. 12).

**Histopathology**. Seeds were imbibed in distilled water for 2 h with gently shaking and a small hole was cut in the seed coat with a surgical scalpel to aid resin infiltration. Seeds were fixed in 4% formaldehyde (Electron Microscopy Sciences) in 50 mM PIPES buffer (pH 7) with gently pulling under vacuum for 10 min. Seeds were left in fixative overnight at 4 °C, dehydrated in an ethanol gradient, and infiltrated with Technovit 7100 plastic resin (Electron Microscopy Sciences) according to manufacturer's instructions. Resin was infiltrated under gentle vacuum for 20 min followed by rotation at 4 °C overnight. Seeds were embedded in beam capsules and 4 μm thick sections were cut on an MR2 manual rotary microtome (RMC Boeckeler) using a glass knife. Sections were stained with 0.02% (w/v) Toluidine blue-O for 30 s, then rinsed and mounted in distilled water. Images were captured on a Leica DM6B microscope (Leica Biosystems Inc. Buffalo Grove, IL) equipped with a Leica DMC 4500 color camera using Leica Application Suite X (LASX) software.

**Bioinformatics**. For mining of DUF1338 homologs in plants all the proteins from completed genomes with protein predictions available by september 2019 were retrieved from the GenBank FTP site. The homologs searched in each pro-teome with BlastP using Flo7 as query with a bit score cutoff of 150 and an e-value cutoff of E-12. The retrieved sequences were aligned using muscle[56], and trimmed using jalview[57]. The multiple sequence alignment was used for phylogenetic reconstruction suing IQ tree, the best amino acid substitution model was selected with ModelFinder implemented in IQtree[58], branch support was calculated using 10000 bootstrap generations

Sequences of DUF1338 homologs were downloaded from Pfam (https://pfam.xfam.org/family/PF07063). To compare the 3D structure of HglS with other protein structures we used VAST[10]. All alignments were done using the MAFFT-LINSI algorithm[59], and alignments were compared with secondary structures and visualized using Easy Sequencing in PostScrip (http://espript.ibcp.fr)[60]. Molecular graphics and analyses were performed with UCSF Chimera, developed by the Resource for Biocomputing, Visualization, and Informatics at the University of California, San Francisco, with support from NIH P41-GM103311[52]. Python scripts were developed to calculate conservation of "HHE" metal-binding triads and R74 residues. Calculation of Shannon-Entropy for DUF1338 sequence conservation was carried out using the python library Protein Dynamics and Sequence Analysis (ProDy)[61,62]. Co-expression analysis of *A. thalina* transcripts was performed using the ATTED-II database[63].

**Reporting summary**. Further information on research design is available in the Nature Research Reporting Summary linked to this article.

## Data availability
The atomic coordinates and structural factors of HglS without substrate, HglS in complex with substrate, and Flo7 in complex with substrate have been deposited in the Worldwide Protein Data Bank (https://www.wwpdb.org/) with PDB ID codes of 6W1G, 6W1H, and 6W1K, respectively. Bacterial strains and plasmids are available upon request from: https://registry.jbei.org/. The source data underlying Figs. 1d, g, h and Supplementary Figs. 3, 4a, b, 5b, and 10c are provided as a source data file. Source data are provided with this paper.

## Code availability
All code used in data analysis will be made available upon request. Source data are provided with this paper.

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

## Acknowledgements

We would like to thank Johan Jaenisch for generously providing *A. thaliana* cDNA. Python code to analyze kinetics data was provided by Sam Curran. This work was part of the DOE Joint BioEnergy Institute (https://www.jbei.org) supported by the US Department of Energy, Office of Science, Office of Biological and Environmental Research, and was part of the Agile BioFoundry (http://agilebiofoundry.org) supported by the US Department of Energy, Energy Efficiency and Renewable Energy, Bioenergy Technologies Office, through contract DE-AC02-05CH11231 between Lawrence Berkeley National Laboratory and the US Department of Energy. J.M.B.H. was supported by the National Science Foundation Graduate Research Fellowship Program under Grant No. DGE 1106400. The Advanced Light Source is a Department of Energy Office of Science User Facility under Contract No. DE-AC02-05CH11231. The Berkeley Center for Structural Biology is supported in part by the Howard Hughes Medical Institute. The ALS-ENABLE beamlines are supported in part by the National Institutes of Health, National Institute of General Medical Sciences, grant P30 GM124169. Use of the Stanford Synchrotron Radiation Lightsource, SLAC National Accelerator Laboratory, is supported by the US Department of Energy, Office of Science, Office of Basic Energy Sciences under Contract No. DE-AC02-76SF00515. The SSRL Structural Molecular Biology Program is supported by the DOE Office of Biological and Environmental Research, and by the National Institutes of Health, National Institute of General Medical Sciences (P41GM103393). The contents of this publication are solely the responsibility of the authors and do not necessarily represent the official views of NIGMS or NIH. The views and opinions of the authors expressed herein do not necessarily state or reflect those of the United States Government or any agency thereof. Neither the United States Government nor any agency thereof, nor any of their employees, makes any warranty, expressed or implied, or assumes any legal liability or responsibility for the accuracy, completeness, or usefulness of any information, apparatus, product, or process disclosed, or represents that its use would not infringe privately owned rights.

## Author contributions

Conceptualization, M.G.T., J.M.B.H., and J.H.P.; Methodology, M.G.T., J.M.B.H., J.H.P., J.F.B., P.C.M., E.E.K.B., J.A.H., W.M.M., M.S.B., and Y.L.; Investigation, M.G.T., J.M.B.H., J.H.P., V.T.B., J.A.H., M.S.B., W.M.M., L.J.W., T.P.B., W.S., R.W.H., C.B.E., E.E.K.B., and Y.L.; Writing – Original Draft, M.G.T, J.M.B.H., and J.H.P.; Writing – Review and Editing, All authors; Resources and supervision H.V.S., P.M.S., M.A.M., P.D.A., and J.D.K.

## Competing interests

J.D.K. has financial interests in Amyris, Lygos, Demetrix, Napigen, Maple Bio, Apertor Labs and Ansa Biotechnology. All other authors declare no competing interests.
