## [Peer Review File · Nature Communications]

Reviewer Comments

Reviewer #1 (Remarks to the Author):

In this manuscript, the authors describe the structural and mechanistic characterization of a Fe(II)-dependent oxygenase, hydroxyglutarate synthase (HglS) that catalyzes the decarboxylation and intramolecular hydroxylation of 2-oxoadipate (2OA) to produce D-2-hydroxyglutarate (D2HG) in the D-lysine catabolic pathway of *Pseudomonas putida*. Using the vector Alignment Search Tool (VAST), the authors show that the structure of HglS is closely related to that of HMS and HPPD, revealing that HglS is likely a member of the vicinal oxygen chelate (VOC) enzyme family. With this knowledge, the authors tested a proposed catalytic mechanism of HglS by measuring O₂ consumption and monitoring ¹⁸O-labeling in vitro. They also used structural data to identify key active-site residues involved in substrate-binding (Arg74, Val402, Ser403), confirmed by assaying enzyme mutants and testing substrate promiscuity. The active-site arginine residue is shown to be highly conserved across DUF1338 homologs across bacteria and eukaryotes. The biochemical function of converting 2OA to 2HG is also shown to be conserved in the homologs found in *E. coli* (YdcJ), *A. thaliana* (AT1G07040), and *O. sativa* (FLO7). Furthermore, plant phenotypic study using the *Arabidopsis* AT1G07040 mutant revealed defect in germination and growth, along with altered morphology of the oil body, similar than that of previously characterized rice FLO7 mutants.

The research is thoroughly conducted and clearly presented. The findings of this study are significant contributions to the area of lysine catabolism in both bacteria and plants. They also expand the knowledge of the two-substrate α -ketoacid-dependent oxygenase family. I only have a few minor comments.

Minor comments:

- Line 91 – make sure the Å is correctly used and shown in the manuscript
- Figure S7 – suggestion to label the enzyme names on the lysine catabolism scheme instead of numbers
- Figure S10 and S11 are missing descriptions. Please provide more information regarding these figures.

Jing-Ke Weng

Reviewer #2 (Remarks to the Author):

In this manuscript by Thompson et al., the structure and natural context of the enzyme HglS (2-oxoadipate dioxygenase) is described. The evidence presented convincingly shows that HglS is the third member of the two-substrate α -ketoacid dependent dioxygenase enzymes (in addition to hydroxyphenylpyruvate dioxygenase and hydroxymandelate synthase). The structure reveals a core fold similar to the other members of this family, but with a quite different domain structure. The structures presented show both the ligand free enzyme reconstituted with nickel and the nickel oxoadipate complex structure as a facsimile of the ES complex. It is also shown that the oxoadipate and O₂ stoichiometry is ~1:1 and that both atoms of oxygen added to from the product arise from dioxygen. The authors expand the work by identifying numerous other proteins from multiple kingdoms that have the salient characteristics of HglS and so indicate that the activity reported is likely common to numerous organisms. It is my assessment that this is a unique and important addition to the catalog of α -ketoacid dioxygenases. The paper is well-written with good adherence to conventional English usage and for the most part well presented. I have only a couple of minor suggestions/ corrections.

Page 5, line 6 – “...assigned the bound metal ion as predominantly Ni(II) with.....” - Nickel is not

the cofactor for this enzyme.

Page 6, middle & in the supplemental – Figure S4 is not labelled with A or B.

Figure 7. I cannot see the point in mutating 402 or 403. Simply knocking out activity doesn't prove much. Conservative mutations may have been more useful. I would suggest removing this section of the manuscript as it is not enlightening.

Note – dioxygen electrode assays of ferrous iron dependent enzymes benefit from the addition of a reductant. The holoenzyme will often inactivate by oxidation with a minute.

Graham R. Moran

Reviewer #3 (Remarks to the Author):

This manuscript entitled "An iron (II) dependent oxygenase performs the last missing step of plant lysine catabolism" identifies the catalytic mechanism of hydroxyglutarate synthase, a member of the previously uncharacterized DUF1338 protein family. The authors show that plant DUF1338 homologs from *Arabidopsis thaliana* and *Oryza sativa* catalyze the conversion of 2-oxoadipate to D-2-hydroxyglutarate in vitro. This reaction step in the postulated lysine catabolic pathway was previously unknown. The findings are novel and highly interesting in terms of both aspects, the unusual biochemical reaction mechanism of oxidative decarboxylation and hydroxylation of a 2-oxoacid and also the metabolic pathway of lysine degradation in plants, which is different from that found in animals. Increasing the level of the essential amino acid lysine in plant-based food is relevant for human and animal nutrition. Lysine metabolism is also closely associated with plant stress response, and several of the catabolic enzymes including the newly identified hydroxyglutarate synthase strongly respond to abiotic as well as biotic stress on a transcriptional level. In general, the work is convincing and will considerably advance knowledge in the field of amino acid catabolism. As far as I can tell (I am not an expert in crystallization and X-ray data analysis) the level of detail provided is sufficient for reproducing the work.

The only aspect that, in my opinion, needs more evidence to strengthen the conclusions is the physiological role of hydroxyglutarate synthase in plant metabolism:

1. T-DNA insertion lines can have more than one insertion site. Thus it is usually expected to check, whether the same effects can be seen in two independent insertion lines or whether the phenotype can be rescued by complementation with the gene of interest. In this study one could argue that two different mutant lines in *Arabidopsis* and rice have been used.
2. The effect of a T-DNA insertion can range from no decrease in gene expression to complete knockout so that not every homozygous line automatically results in "disruption" of the target gene. Therefore, it is important to quantify the effect of the T-DNA insertion in a given line ideally by activity tests or alternatively by expression analysis or protein quantification.
3. The germination phenotype alone is quite unspecific and can be the consequence of many disturbances in metabolism. To provide clear evidence for a role in plant lysine catabolism it is required to analyze metabolite levels in the mutant lines and show whether lysine and/or intermediates of its degradation pathway accumulate (which is the case in e.g. D2HGDH (Araújo et al. 2010, doi: 10.1105/tpc.110.075630) and LKR/SDH (Zhu et al. 2001, doi: 10.1104/pp.126.4.1539) mutant lines). Lysine accumulation in the leaves can be stimulated by extended darkness, which leads to carbohydrate starvation and thus induces protein and amino acid catabolism.

I would suggest to move the paragraph describing the biochemical characteristics of plant

hydroxyglutarate synthases (pages 14-15) up to page 11 and conclude with the part addressing the potential role in plant metabolism, which will be a good starting point for further studies.

Best regards,
Tatjana Hildebrandt

Reviewer #1 (Remarks to the Author):

In this manuscript, the authors describe the structural and mechanistic characterization of a Fe(II)-dependent oxygenase, hydroxyglutarate synthase (HglS) that catalyzes the decarboxylation and intramolecular hydroxylation of 2-oxoadipate (2OA) to produce D-2-hydroxyglutarate (D2HG) in the D-lysine catabolic pathway of *Pseudomonas putida*. Using the vector Alignment Search Tool (VAST), the authors show that the structure of HglS is closely related to that of HMS and HPPD, revealing that HglS is likely a member of the vicinal oxygen chelate (VOC) enzyme family. With this knowledge, the authors tested a proposed catalytic mechanism of HglS by measuring O₂ consumption and monitoring ¹⁸O-labeling in vitro. They also used structural data to identify key active-site residues involved in substrate-binding (Arg74, Val402, Ser403), confirmed by assaying enzyme mutants and testing substrate promiscuity. The active-site arginine residue is shown to be highly conserved across DUF1338 homologs across bacteria and eukaryotes. The biochemical function of converting 2OA to 2HG is also shown to be conserved in the homologs found in *E. coli* (YdcJ), *A. thaliana* (AT1G07040), and *O. sativa* (FLO7). Furthermore, plant phenotypic study using the *Arabidopsis* AT1G07040 mutant revealed defect in germination and growth, along with altered morphology of the oil body, similar than that of previously characterized rice FLO7 mutants.

The research is thoroughly conducted and clearly presented. The findings of this study are significant contributions to the area of lysine catabolism in both bacteria and plants. They also expand the knowledge of the two-substrate α -ketoacid-dependent oxygenase family. I only have a few minor comments.

Minor comments:

- Line 91 – make sure the Å is correctly used and shown in the manuscript

Å has now been used correctly in the manuscript.

- Figure S7 – suggestion to label the enzyme names on the lysine catabolism scheme instead of numbers

We have now labelled the enzymes in the scheme.

- Figure S10 and S11 are missing descriptions. Please provide more information regarding these figures.

We have now provided more information for these figures.

Jing-Ke Weng

Reviewer #2 (Remarks to the Author):

In this manuscript by Thompson et al., the structure and natural context of the enzyme HglS (2-oxoadipate dioxygenase) is described. The evidence presented convincingly shows that HglS is the third member of the two-substrate alpha-ketoacid dependent dioxygenase enzymes (in addition to hydroxyphenylpyruvate dioxygenase and hydroxymandelate synthase). The structure reveals a core fold similar to the other members of this family, but with a quite different domain structure. The structures presented show both the ligand free enzyme reconstituted with nickel and the nickel oxoadipate complex structure as a facsimile of the ES complex. It is also shown that the oxoadipate and O₂ stoichiometry is ~1:1 and that both atoms of oxygen added to from the product arise from dioxygen. The authors expand the work by identifying numerous other proteins from multiple kingdoms that have the salient characteristics of HglS and so indicate that the activity reported is likely common to numerous organisms. It is my assessment that this is a unique and important addition to the catalog of alpha-ketoacid dioxygenases. The paper is well-written with good adherence to conventional English usage and for the most part well presented. I have only a couple of minor suggestions/ corrections.

Page 5, line 6 – “...assigned the bound metal ion as predominantly Ni(II) with.....”
- Nickel is not the cofactor for this enzyme.

We have revised the text so that it now reads “...assigned the co-crystallized metal as...”

Page 6, middle & in the supplemental – Figure S4 is not labelled with A or B

We have now labelled the left panel of Figure S4 A and the right panel B.

Figure 7. I cannot see the point in mutating 402 or 403. Simply knocking out activity doesn't prove much. Conservative mutations may have been more useful. I would suggest removing this section of the manuscript as it is not enlightening.

We feel that the language in the manuscript accurately reflects the reviewer's comment that the V402P mutation does not prove much. We state that “Val402 significantly decreased enzyme activity, suggesting that while this residue is not essential for turnover, it likely contributes to substrate binding affinity.” We have

not removed this sentence from the manuscript but instead have added a sentence at the end of this paragraph to address this topic further. The sentence reads "Further experiments with more conservative mutations at positions 402 and 403 will be required to fully understand the role(s) of these residues in substrate binding and/or catalysis."

Note – dioxygen electrode assays of ferrous iron dependent enzymes benefit from the addition of a reductant. The holoenzyme will often inactivate by oxidation with a minute.

Graham R. Moran

Reviewer #3 (Remarks to the Author):

This manuscript entitled "An iron (II) dependent oxygenase performs the last missing step of plant lysine catabolism" identifies the catalytic mechanism of hydroxyglutarate synthase, a member of the previously uncharacterized DUF1338 protein family. The authors show that plant DUF1338 homologs from *Arabidopsis thaliana* and *Oryza sativa* catalyze the conversion of 2-oxoadipate to D-2-hydroxyglutarate in vitro. This reaction step in the postulated lysine catabolic pathway was previously unknown. The findings are novel and highly interesting in terms of both aspects, the unusual biochemical reaction mechanism of oxidative decarboxylation and hydroxylation of a 2-oxoacid and also the metabolic pathway of lysine degradation in plants, which is different from that found in animals. Increasing the level of the essential amino acid lysine in plant-based food is relevant for human and animal nutrition. Lysine metabolism is also closely associated with plant stress response, and several of the catabolic enzymes including the newly identified hydroxyglutarate synthase strongly respond to abiotic as well as biotic stress on a transcriptional level. In general, the work is convincing and will considerably advance knowledge in the field of amino acid catabolism. As far as I can tell (I am not an expert in crystallization and X-ray data analysis) the level of detail provided is sufficient for reproducing the work.

The only aspect that, in my opinion, needs more evidence to strengthen the conclusions is the physiological role of hydroxyglutarate synthase in plant metabolism:

1. T-DNA insertion lines can have more than one insertion site. Thus it is usually expected to check, whether the same effects can be seen in two independent insertion lines or whether the phenotype can be rescued by complementation with

the gene of interest. In this study one could argue that two different mutant lines in Arabidopsis and rice have been used.

2. The effect of a T-DNA insertion can range from no decrease in gene expression to complete knockout so that not every homozygous line automatically results in “disruption” of the target gene. Therefore, it is important to quantify the effect of the T-DNA insertion in a given line ideally by activity tests or alternatively by expression analysis or protein quantification.

3. The germination phenotype alone is quite unspecific and can be the consequence of many disturbances in metabolism. To provide clear evidence for a role in plant lysine catabolism it is required to analyze metabolite levels in the mutant lines and show whether lysine and/or intermediates of its degradation pathway accumulate (which is the case in e.g. D2HGDH (Araújo et al. 2010, doi: 10.1105/tpc.110.075630) and LKR/SDH (Zhu et al. 2001, doi: 10.1104/pp.126.4.1539) mutant lines). Lysine accumulation in the leaves can be stimulated by extended darkness, which leads to carbohydrate starvation and thus induces protein and amino acid catabolism.

We thank the reviewer for these suggestions but we believe that these experiments may be outside the scope of the paper, given the breadth of biochemical, structural, and enzymatic evidence that we present in the manuscript as well as the breadth of evidence that support our claims in the literature.

The central claim in our paper is that HglS catalyzes the last gap in plant lysine metabolism which is known to proceed up to 2-oxoadipate. In our work we show conclusively with biochemical studies that two distantly related plant enzymes (*A. thaliana* and *O. sativa*) catalyze the conversion of 2-oxoadipate to 2-hydroxyglutarate. While 2-hydroxyglutarate is an established intermediate in plant lysine catabolism, no rigorous biochemical proof existed for its formation. Our work succinctly draws a direct link between these intermediates. Our claims are supported by the high correlation of HglS expression in *A. thaliana* to the other known pathway enzymes, and wide distribution of these enzymes across nearly all plant species which other groups have also observed.

Authors of published work that describe the physiological role of the HglS homolog in rice FLO7, noted the similarity of the *floury* phenotype of FLO7 mutants to lysine accumulating *opaque2* mutants of maize. Though these authors believed at the time that FLO7 was a transcription factor, we clearly show structurally and biochemically that it is a bona fide HglS. Published work in *A. thaliana* has shown that disruption of lysine catabolism results in a delayed germination phenotype, which is also present in confirmed T-DNA mutant lines of the *A. thaliana* HglS homolog Salk_103299C. In conjunction with the above mentioned biochemical

evidence, we believe that we clearly show that HgIS is the last missing step of plant lysine catabolism, and that further plant physiology investigation is beyond the scope of this manuscript.

I would suggest to move the paragraph describing the biochemical characteristics of plant hydroxyglutarate synthases (pages 14-15) up to page 11 and conclude with the part addressing the potential role in plant metabolism, which will be a good starting point for further studies.

We agree with the reviewer and have now moved the biochemical and structural work on the plant homologs onto page 11. We believe that this structural change better highlights the importance of establishing the biochemical connection between 2OA and 2HG in plants, and indeed serves as a better launching point for further work.